# The Hidden Story of Heterogeneous B-raf V600E Mutation Quantitative Protein Expression in Metastatic Melanoma—Association with Clinical Outcome and Tumor Phenotypes

**DOI:** 10.3390/cancers11121981

**Published:** 2019-12-09

**Authors:** Lazaro Hiram Betancourt, A. Marcell Szasz, Magdalena Kuras, Jimmy Rodriguez Murillo, Yutaka Sugihara, Indira Pla, Zsolt Horvath, Krzysztof Pawłowski, Melinda Rezeli, Kenichi Miharada, Jeovanis Gil, Jonatan Eriksson, Roger Appelqvist, Tasso Miliotis, Bo Baldetorp, Christian Ingvar, Håkan Olsson, Lotta Lundgren, Peter Horvatovich, Charlotte Welinder, Elisabet Wieslander, Ho Jeong Kwon, Johan Malm, Istvan Balazs Nemeth, Göran Jönsson, David Fenyö, Aniel Sanchez, György Marko-Varga

**Affiliations:** 1Clinical Protein Science & Imaging, Biomedical Centre, Department of Biomedical, Engineering, Lund University, BMC D13, 221 84 Lund, Sweden; lazaro.betancourt@med.lu.se (L.H.B.); hzsmisi@gmail.com (Z.H.); melinda.rezeli@bme.lth.se (M.R.); jeovanis.gil_valdes@med.lu.se (J.G.); jonatan.eriksson@bme.lth.se (J.E.); roger.appelqvist@bme.lth.se (R.A.); gyorgy.marko-varga@bme.lth.se (G.M.-V.); 2Cancer Center, Semmelweis University, Budapest 1083, Hungary; 3Section for Clinical Chemistry, Department of Translational Medicine, Lund University, Skåne University Hospital Malmö, 205 02 Malmö, Sweden; magdalena.kuras@med.lu.se (M.K.); indira.pla_parada@med.lu.se (I.P.); krzysztof_pawlowski@sggw.pl (K.P.); johan.malm@med.lu.se (J.M.); aniel.sanchez@med.lu.se (A.S.); 4Division of Physiological Chemistry I, Department of Medical Biochemistry and Biophysics, Karolinska Institutet, SE-17 177 Stockholm, Sweden; jimmy.esneider_rodriguez@med.lu.se(J.R.M.); yutaka.sugihara@med.lu.se (Y.S.); 5Department of Biochemistry and Microbiology, Warsaw University of Life Sciences, 02-787 Warsaw, Poland; 6Department of Molecular Medicine and Gene Therapy, Lund Stem Cell Center, Lund University, BMC A12, Sölvegatan 17, 221 84 Lund, Sweden; kenichi.miharada@med.lu.se; 7Translational Science, Cardiovascular Renal and Metabolism, IMED Biotech Unit, AstraZeneca, 431 50 Gothenburg, Sweden; Tasso.Miliotis@astrazeneca.com; 8Division of Oncology and Pathology, Department of Clinical Sciences Lund, Lund University, 221 85 Lund, Sweden; bo.baldetorp@med.lu.se (B.B.); hakan.olsson@med.lu.se (H.O.); lotta.lundgren@med.lu.se (L.L.); charlotte.welinder@med.lu.se (C.W.); elisabet.wieslander@med.lu.se (E.W.); goran_b.jonsson@med.lu.se (G.J.); 9Department of Surgery, Clinical Sciences, Lund University, Skåne University Hospital, 222 42 Lund, Sweden; christian.ingvar@med.lu.se; 10Department of Analytical Biochemistry, Faculty of Science and Engineering, University of Groningen, 9712 CP Groningen, The Netherlands; peter.horvatovich@gmail.com; 11Department of Biotechnology, Yonsei University, Seoul 03722, Korea; kwonhj@yonsei.ac.kr; 12Department of Dermatology and Allergology, University of Szeged, H-6720 Szeged, Hungary; nemeth.istvan.balazs@med.u-szeged.hu; 13Institute for Systems Genetics, NYU School of Medicine, 550 1st Ave, New York, NY 10016, USA; david@fenyolab.org

**Keywords:** malignant melanoma, BRAF V600E mutation, proteomics, mass spectrometry genetics, heterogeneity, prognosis

## Abstract

In comparison to other human cancer types, malignant melanoma exhibits the greatest amount of heterogeneity. After DNA-based detection of the BRAF V600E mutation in melanoma patients, targeted inhibitor treatment is the current recommendation. This approach, however, does not take the abundance of the therapeutic target, i.e., the B-raf V600E protein, into consideration. As shown by immunohistochemistry, the protein expression profiles of metastatic melanomas clearly reveal the existence of inter- and intra-tumor variability. Nevertheless, the technique is only semi-quantitative. To quantitate the mutant protein there is a fundamental need for more precise techniques that are aimed at defining the currently non-existent link between the levels of the target protein and subsequent drug efficacy. Using cutting-edge mass spectrometry combined with DNA and mRNA sequencing, the mutated B-raf protein within metastatic tumors was quantitated for the first time. B-raf V600E protein analysis revealed a subjacent layer of heterogeneity for mutation-positive metastatic melanomas. These were characterized into two distinct groups with different tumor morphologies, protein profiles and patient clinical outcomes. This study provides evidence that a higher level of expression in the mutated protein is associated with a more aggressive tumor progression. Our study design, comprised of surgical isolation of tumors, histopathological characterization, tissue biobanking, and protein analysis, may enable the eventual delineation of patient responders/non-responders and subsequent therapy for malignant melanoma.

## 1. Introduction

Malignant melanoma (MM) is one of the most aggressive and heterogeneous of all human cancer types [1]. The melanoma subtypes differ in origin, location and mutational profile [2]. At the genetic level, BRAFmut, RASmut, NF1mut, and triple WT subgroups exist; while, at the transcriptomic level, low and high immune cell-infiltrated prognostic subtypes have emerged. These are reflected in the pathological categorization of brisk and non-brisk patterns of tumor-infiltrating lymphocytes [3]. With respect to the variation in clinical symptoms, appearance, and the biology of patients, combined with the morphological and molecular variation of an individual tumor; malignant melanoma is one of the most heterogeneous of all diseases [4].

Most cases of malignant melanoma are diagnosed at an early stage, where surgical excision is curative [5]. The management of patients with disseminated disease, however, is troublesome. For instance, checkpoint immunotherapy with monoclonal antibodies targeting CTLA-4 and PD-1 has provided clinically-important benefits for only a subset of melanoma patients. This treatment modality was developed in parallel with targeted mitogen-activated protein kinase (MAPK) pathway inhibitor therapies, such as vemurafenib and dabrafenib, alone, or when combined with trametinib and cobimetinib [6,7,8]. In BRAF V600E-mutated melanomas, such therapeutical approaches inhibit the activity of key members of the MAPK pathway, such as BRAF and MEK. The treatment has resulted in a significantly higher response rate in reducing the bulky tumor mass, however, efficacy still varies, and after a period free from disease advancement, most responsive patients develop resistance to the therapy and lethally progress [9].

In the clinical setting, the gold standard are FDA-approved PCR-based DNA tests that selectively amplify the mutant BRAF gene [10]. These tests reveal the presence or absence of a specific BRAF gene mutation; however, there is no indication as to whether the BRAF gene is transcribed and translated into the B-raf protein. Therefore, it is still unclear whether BRAF regulates expression at the mRNA level or the protein level. The therapeutic target of the clinically-administered drugs is the protein. Therefore, it is important to evaluate the relationship between drug efficacy and protein expression level. Thus, there is a current lack of crucial information that could aid clinical decisions.

Proteomics is a highly-promising field of research that can assist in identifying therapeutic targets and cancer biomarkers [11,12,13,14,15]. At the core of proteomics is the technique of mass spectrometry (MS) that provides a sensitive analysis of complex mixtures of proteins and peptides. In addition, MS provides a means of undertaking unsurpassed challenges that exist in genomics; including protein identification, studying post-translational modifications, and determining the relative abundance of protein products [16].

The identification and quantitation of B-raf V600E protein by mass spectrometry is challenging. To date, only one study has attempted to address the issues of heterogeneity that are inherent to the expression of WT and V600E BRAF [17]. Using immunoenrichment-based techniques following a targeted liquid chromatography multiple reaction monitoring (LC-MRM) approach, both proteins were identified and quantitated in complex biological samples comprised of colorectal carcinoma CRC cell lines and tissue specimens. Thus, identification and quantitation by mass spectrometry of the V600E B-raf protein remains an understudied topic for most cancerous tissues, including melanoma.

In the current work, we summarize a group of observations made on small set of samples that emphasizes the importance of the analysis of the B-raf V600E protein on melanoma metastases. We were able to identify and quantify, for the first time, the mutated protein by mass spectrometry. The samples were also screened for the mutation at the DNA and mRNA level. Our study revealed a subjacent layer of heterogeneity in V600E BRAF-mutated melanomas. Two distinct groups associated with different clinical outcomes and molecular features were apparent. The data indicated that patients with tumors displaying a higher level of B-raf V600E expression had a poorer prognosis than patients with a lower level of the mutated protein. For both tumor groups, morphological differences in histological images and protein differential expression profiles supported the novel findings of distinctive tumor phenotypes.

When combined with optimized, high-quality biobanking protocols and technology-driven protein analytical approaches, our results illustrated that surgically-removed and histologically well-characterized tumor tissues can provide important new insights into the expression of key protein mutations. Validation of our results in a larger cohort of metastatic melanoma patients will support our conclusions and could be beneficial for the treatment of melanoma patients. Targeted inhibitor therapies are currently directed towards the protein and not the gene. Thus, such unique information may ultimately impact the management of advanced-stage melanoma.

## 2. Results

### 2.1. B-raf V600E Mutant Protein Expression Is a Heterogeneous Event

Although DNA and RNA sequencing indicate the presence of mutated BRAF, there is no obvious correlation between this information and the inter- and intra-tumoral abundance and distribution of the mutant protein. With immunohistochemistry it is possible to map the distribution of B-raf V600E within the tumor (Figure 1). The first example shows a tumor with homogeneous expression of the mutant protein throughout the tumor tissue (Figure 1A). Conversely, an example of intra-tumor heterogeneity is illustrated by two different regions within the same tumor (Figure 1B). This example highlights the pathological heterogeneous expression pattern that can eventuate for the mutated B-raf V600E protein (using the anti-BRAF V600E (VE1) monoclonal antibody), and is epitomized by highly irregular cellular assignment.

In routine daily practice, conventional immunohistochemistry is readily available to identify the mutated B-raf V600E protein. With the VE1 clone, a fairly specific staining can be observed that is comparable to genetic analysis; however, there are no accurate cut-off values for cases with positive heterogeneity. Therefore, there are still some instances where there is limited potential for accurate microscopic assessment, e.g., when only focal sparse positivity is observed, or when intra-tumoral heterogeneity is apparent with more pronounced, focal B-raf V600E expression (Figure 1B, dashed lines). Due to the intrinsic limitations of the technique, inter-tumor comparison of mutated protein levels is only qualitative. Mass spectrometry not only provides a means of discriminating between B-raf WT and V600E, but, also, to accurately quantitate the proteins.

### 2.2. B-raf V600E Mutant and B-raf WT Protein Identification

The BRAF mutational status was studied in 56 tumor samples from metastatic melanoma patients. Tumors were firstly characterized by histopathology and frozen sections were analyzed using a semi-automated proteomic workflow (Materials and Methods and Figure 2). The methodology is comprised of automatic protein extraction from tumors; the automated denaturation and tryptic digestion of proteins on a robotic micro-chromatographic platform; peptide labeling with TMT 11-plex reagents, HpH RP-HPLC fractionation, LC-MS/MS analysis of collected fractions; and finally, protein identification and quantitation.

More than 12,000 proteins were identified and quantified with this strategy, and nearly 9000 were common to the six batches of TMT 11-plex experiments. This included the mutated B-raf V600E, which was identified and quantitated for the first time by mass spectrometry in malignant melanoma tissue samples. Corresponding to the simultaneous analysis of ten MM samples, Figure 3 shows the assigned MS/MS spectra of TMT-labeled peptides for the mutated B-raf (IGDFGLATEK, Figure 3A) and the WT protein (IGDFGLATVK, Figure 3B).

The BRAF mutational status of 52 of the 56 samples was previously determined at DNA and mRNA levels. The results obtained were in agreement for 50 of the cases (Appendix A). The mRNA study, however, was performed on the same region of the tumor that had been selected for proteomics and detected a higher number of samples containing the BRAF V600E mutation; thus, this data provided a more valuable measure against which the mass spectrometry results could be compared.

Twenty B-raf V600E positive tumors were found by proteomics and, except for two instances, were in agreement with the mRNA-based study (Table 1 and Appendix A). The DNA-based analysis of the discordant cases confirmed the results of the RNA-based study and the samples were excluded from the group of metastases expressing the mutation. In addition, the WT B-raf protein was observed in all tumor samples. Overall, the proteomic results aligned with the data obtained by genomics in 50 of the 52 samples.

### 2.3. B-raf V600E Expression and Correlation with Patient Survival and Tumor Phenotype

The clinical data from the metastatic melanoma patients and the BRAF status from genomic and mass spectrometric determination are combined in Figure 4A. Noticeably, this heat map representation captured the limited information provided by both genomic studies regarding BRAF V600E mutation, compared to the more complex picture offered by the protein expression. As determined by mass spectrometry, the relative abundance of the B-raf V600E protein revealed a high degree of variability with a coefficient of variation of 57% across the twenty mutation-positive metastatic melanomas that were also verified at mRNA level. Some metastatic tissues had high levels of the mutant protein and appeared to be responsible for most of the observed variability. Other tissues had lower, but similar, levels of mutant protein expression (Figure 4B).

We investigated the correlation between the relative abundance of the B-raf V600E protein and patient survival. We noticed a marked imbalance towards a higher overall survival of patients younger than 40 years old for both BRAF V600E positive tumor and the whole sample set (Appendix A), suggesting a different progression of melanoma for this group of patients. Several studies have also shown that survival is higher below this age [18,19,20]. Consequently, four patients <40 years of age at diagnosis were excluded from the analysis. The data revealed that B-raf V600E protein expression is significantly negatively correlated with patients’ overall survival (r = −0.58, *p* = 0.048). Univariate analysis generated two groups of patients with distinct differences in survival and significantly reduced survival was associated with high expression of the B-raf V600E mutated protein (Figure 5A). The median overall survival for the two groups was markedly different; 248 days for the nine patients with the highest B-raf mutation levels and 2460 days for the seven patients with a lower expression of the B-raf mutation. Notably, all patients with high levels of B-raf V600E-expressing tumors did not survive beyond 18 months. This result suggests that protein expression of the B-raf V600E mutation in the tumor could be a significant risk factor for poorer prognosis of patients <40 years of age with stage 3/4 malignant melanoma.

Next, histological images of mutation-positive metastatic melanoma samples were examined to determine if any apparent morphological relationships exist between the high and low B-raf V600E mutant-expressing groups (Figure 5B and Appendix A). For tumors that expressed high levels of B-raf V600E, an increased vascularization was apparent. In addition, the cells were generally smaller but heterogeneous in size and had a non-cohesive pattern (Figure 5B, a and b images). Conversely, the cells from tumors with a lower expression of the B-raf V600E protein were less heterogeneous. This group was comprised of larger cells that often displayed multinucleation, a deeper cytoplasmic color, cell grouping, and connective tissue septa (Figure 5B, c and d images). Based on the above-described features, a heterogeneity score (0–4) was calculated. The total score was equal to the tumor cell size variation + vascularization + discohesion + multinucleation. In order to accept a feature for the group, >55% of cases had to display a specific property (Table 2).

Whether the expression of the BRAF V600E mutation could define distinct molecular phenotypes for the two groups of mutation-positive metastatic melanomas was also explored. The 697 differentially-expressed proteins, determined from the high and low B-raf V600E-expressing tumors, were used to direct a hierarchical clustering at the level of protein quantitation. As expected, clear differences in protein abundance were apparent with the data dividing into two major clusters corresponding to the two groups (Figure 5C). This was also evident in the PCA plot (Figure 5D). A strong discrimination between high and low B-raf V600E-expressing tumors was observed in both analyses.

### 2.4. Protein Profiles Associated with B-raf V600E Expression

Functional analysis of the 697 differentially-expressed proteins with the Ingenuity IPA system provided interesting insights. The overrepresented molecular functions included RNA post-transcriptional modification, cell-to-cell signaling and interaction, cellular development and cellular growth and proliferation. The top canonical pathways included B cell development, EIF2 signaling, antigen presentation pathway, B cell receptor signaling, natural killer cell signaling, and actin cytoskeleton signaling (see Figure 6).

When the IPA was performed only on the proteins that were up- and down-regulated in samples with a high expression of B-raf V600E, a distinct picture emerged. The down-regulated proteins in the B-raf V600E_high samples were significantly related to inflammatory response, cell-to-cell signaling and interaction, cellular growth and proliferation. Up-regulated proteins in the B-raf V600E_high samples were significantly related to RNA post-transcriptional modification, protein synthesis, gene expression and cell death and survival.

IPA-derived relational networks reflect the possible roles of known oncogenes, e.g., MYC and HDGF (REF: PMID 27543492), overexpressed in B-raf V600E_high samples (Figure 6C), and tumor suppressors, e.g., PML and toll-like receptors (PMID: 30127747, 29416846), downregulated in B-raf V600E_high samples (Figure 6B).

## 3. Discussion

B-raf is a key regulatory protein in malignant melanoma that has a disparate and irregular mechanism of expression. This element is a major factor in understanding melanoma pathobiology. To verify the BRAF mutational status of a tumor, standard molecular pathology procedures are based on DNA sequencing. Diagnosis of BRAF mutations with this approach has not always been successful and sometimes failures in verification have led to major clinical consequences for the patients [21].

To date, direct analysis of the protein in melanoma tumors has been achieved by immunohistochemistry [22,23,24]. In comparison to DNA-based assays, these studies have shown a high sensitivity and specificity in detecting BRAF mutants, with a study reporting B-raf V600E protein expression as a novel prognostic marker in primary melanoma [25]. At present, however, the main limitations of the immunohistochemical approach are related to the possibility of false negatives because of the high degree of heterogeneity, and apparent inter-observer variability [26,27,28].

In this study, the B-raf V600E mutant was identified by mass spectrometry in metastatic melanoma samples for the first time. When compared to the mRNA-based study performed on the same tissue samples, the identification of the B-raf V600E protein achieved 100% sensitivity and 91% specificity (Table 1). Discrepancies in the status of BRAF using different techniques are not uncommon and may reflect the high degree of intra-tumoral heterogeneity often observed with BRAF V600E mutations [29,30,31,32].

WT B-raf was also identified in this study; however, quantitation of the protein remains a challenge. Here, the relative abundance of the protein can only be determined based on unique tryptic peptides that ensured unambiguous quantitation of WT B-raf, RAF-1 and ARAF proteins. These proteins share more than 40% sequence homology. The V600E mutation, however, is located in a region that generates a tryptic peptide that is identical for all three proteins. Therefore, quantitation of the WT B-raf with our current data and experimental design was not attempted.

For the past few decades, the prognostic factors for melanoma in clinical practice have remained unaltered. Pathological staging that primarily focuses on tumor thickness is still applied as an estimate of the clinical behavior of the primary melanoma [33]. The mitotic score at the primary site is also often assessed [34,35]. Nevertheless, as a consequence of the diverse and contradictory information surrounding patient survival when analyzing both primary and metastatic melanomas, the prognostic value of BRAF mutations is still under discussion [25,36,37,38,39,40,41].

Albeit in a small sample set of patients, ≥40 years of age, by focusing the analysis of B-raf V600E on protein quantitation rather than mutational status, our data suggest that the B-raf V600E expression level could be a prognostic factor for metastatic melanoma. Compared to metastatic tissue from patients with lower expression levels of B-raf V600E, higher levels were associated with worse overall survival. Similar observations have been made in an immunohistochemistry study of primary tumors [25]. The authors showed that survival was significantly reduced for patients with strongly positive B-raf V600E expression compared to the group with weakly positive expression. To the best of our knowledge, although at different stages of the disease, that and our study are the only investigations that have analyzed B-raf V600E protein expression and reached the same correlation with clinical outcome. Nevertheless, when the disease is progressing and giving rise to metastases, whether in the lymph nodes or any distant organ, it is subject to evolutional changes [42,43], and it may differ from the primary tumor in certain characteristics, e.g., morphology, tumor mutational burden, and, eventually, BRAF status [44,45]. At the time of the actual treatment, the latest information source is key to the optimal choice of therapy, thus, metastases shall be sampled and analyzed at time of progression [46,47]. Histological images of high- and low-expressing B-raf V600E tumors showed different features, from which a heterogeneity score was calculated. Although the evaluation was performed on a limited number of samples, these features revealed that tumors that expressed higher levels of the B-raf V600E protein had a higher degree of heterogeneity compared to tumors expressing lower levels of B-raf V600E. The later tumor group showed a tendency towards multinuclear and larger cells, characteristics that have been associated with senescence [48,49,50]. Interestingly, a previous study has shown that B-raf V600E-expressing melanocytes display classical hallmarks of senescence, thus suggesting that oncogene-induced senescence represents a genuine protective physiological process [51]. On the contrary, the tendency of the tumor group expressing high levels of B-raf V600E towards neovascularization is indicative of rapid cancer cell proliferation, tumor invasion and cancer progression in general [52].

Most BRAF V600E positive metastases had tumor content >75% (Appendix A). The only exception was sample MM116, with 15% of tumor cells. This was indicative of a high concentration of BRAF V600E in the sample. In fact, MM116 belongs to the group of tumors with high expression of BRAF V600E, matching the distinctive characteristics of this group of metastases.

The protein profiles associated with the high and low levels of B-raf V600E had striking functional features. For the proteins that were significantly more abundant in the B-raf V600E_low samples, there was a clear over-representation of proteins involved in immune response pathways, e.g., PML, toll-like receptors, and HLA antigens [53,54]. Indeed, it is a well-appreciated and known fact that the activation of the immune response correlates with an improved prognosis.

Conversely, proteins that were significantly more abundant in the B-raf V600E_high samples were clearly related to the proliferative nature of these tumors (and poor prognosis) and include several known oncogenes tumor drivers, e.g., the transcription factor MYC [55], and the growth factor HDGF [56]. Also, functional analysis of these proteins highlighted the over-representation of ribosome biogenesis proteins, thus reflecting the proliferative nature of an aggressive tumor type [57]. This preliminary data provides indication of a correlation between the levels of the B-raf V600E protein variant and cancer-related molecular pathways.

The availability of advanced proteomic and high-resolution nano-LC-MS/MS techniques has enabled the deeper mining of melanoma proteome analysis and the determination of the relative abundance of BRAF V600E protein expression across patient tumor metastases. The accuracy of the MS quantitation approach (TMT 11-plex) is still a challenge, as co-isolation interference phenomenon might occur (Appendix A), i.e., where multiple peptides are being selected for MS/MS analysis within the same isolation window.

These effects could impact the evaluation of actual protein abundances to some extent. Hence, further investigation of BRAF V600E protein in melanoma samples should include an acquisition method that effectively eliminates the mentioned issues with TMT 11-plex quantitation. Future experiments should also consider the study of FFPE samples, which is the standard procedure in the clinic for melanoma tissue diagnosis, and the comparison of MS data with a more sensitive technique for BRAF V600E mutation detection and also with other protein-based assays, such as immunohistochemistry.

## 4. Materials and Methods

See Appendix A for details.

### 4.1. Tissue Specimens

The study was approved by the Regional Ethical Committee at Lund University, Southern Sweden, approval numbers: DNR 191/2007, 101/2013 and 2015/266, 2015/618. All patients included in the study provided written, informed consent. The malignant metastatic tissue used in the study was fresh-frozen during surgery at Lund University Hospital and stored in the Melanoma biobank, BioMEL, Region Skåne, Sweden. cDNA direct sequencing of BRAF V600E and the proteomic study were performed on pieces of the tumor tissues. A reference sample was developed to include a broad and representative melanoma metastasis disease presentation from *n* = 40 patient tumors.

### 4.2. Patient Characteristics

A total of 56 patients diagnosed with metastatic melanoma were evaluated in the study (Table 1 and Appendix A). Only two received targeted B-raf treatment with vemurafenib. There were 40 men and 16 women among the investigated cases. Average age ± standard deviation (range) at diagnosis of metastases was 64.1 ± 11.7 (24–89) years. The overall survival was 2.9 ± 3.5 (0.1–17.4) years. The majority of metastatic tissue studied were from the lymph nodes (82%), while the remainder were cutaneous, subcutaneous and visceral. Appendix A details patient clinical data, as well as the tumor content of analyzed samples.

### 4.3. BRAF V600E Mutation Testing

BRAF V600E analysis results at the DNA level were obtained from melanoma patient records stored at Lund University Hospital and were obtained following procedures detailed before [41,58].

Direct sequencing cDNA (ds-cDNA) was performed as previously described [14]. Briefly, mRNA was extracted from frozen melanoma tissue specimens and cDNA was synthesized. Using a pair of BRAF-specific primers, BRAF cDNA was amplified by polymerase chain reaction (PCR). After purification of the PCR products, BRAF mutation status was determined by Sanger sequencing

### 4.4. Protein Extraction, Digestion and Automated C18 Desalting Workflow

The protocols for protein extraction and digestion have been previously described [59]. Protein extraction was performed on sectioned, fresh-frozen human metastatic tissue (10 µm) using the Bioruptor plus, model UCD-300 (Dieagenode). Protein digestion and peptide desalting were performed on the AssayMAP Bravo (Agilent Technologies, Lexington, MA, USA) platform with the urea solution digest and peptide cleanup v2.0 protocols.

### 4.5. TMT 11-plex Labeling and Off Line High pH Fractionation

Peptides were labeled with TMT 11-plex reagents according to the instructions provided by the manufacturer. The reference sample was placed in channel 126 of the TMT 11-plex experiment in order to reduce the effect of isotopic cross-contamination, as a recent study has recommended. [60]. The labeled peptides were mixed together and then purified and concentrated on a C-18 Sep-Pak (Waters Chromatography Europe, Etten-Leur, The Netherlands). TMT-11 labelled peptides were fractionated by high pH RP-HPLC using a Phenomenex Aeris Widepore XB-C8 (3.6 μm, 2.1 × 100 mm) column on an 1100 Series HPLC (Agilent Technologies). Ninety-eight fractions were collected at 1 min intervals and further concatenated to 24 or 25 fractions. This procedure was repeated for the six batches of TMT 11-plex analyzed.

### 4.6. nLC-MS/MS Analysis

nLC-MS/MS analysis of peptide fractions (one-time per fraction) were performed on an Ultimate 3000 HPLC coupled to a Q Exactive HF-X mass spectrometer (Thermo Scientific, San Jose, CA, USA).

### 4.7. Histopathological Evaluation

After each set of slides were submitted for molecular analysis, step-wise sectioning of the tissues was performed. Thus, on average, three sections were evaluated by a board-certified pathologist. Two to three µm thick, frozen tissue sections were placed on glass slides, stained with hematoxylin and eosin, and then placed in an automated slide scanner system (Zeiss Mirax, Jena, Germany). The slides were then evaluated for tissue content, as previously published [61]. Taking into account features that could be further captured based on morphology, all slides from each patient were evaluated and the properties scored as present or not present (0/1). To derive a yield of tissue heterogeneity, a score (0–4) was calculated and incorporated into a sum equaling tumor cell size variation + vascularization + discohesion + multinucleation. To accept a feature for the entire group, >55% of the cases must display the specific property.

### 4.8. Immunohistochemistry

For immunohistochemistry, 4 µm thick, paraffin-embedded tissue sections were firstly placed on silanized slides. An automated immunohistochemical protocol was then performed using a BOND Autostainer and Polymer Refine Detection system (Leica Biosystems Inc., Richmond, IL, USA). Anti-B-raf V600E antibody (clone VE1, Spring Bioscience Corp., Pleasanton, CA, USA) was used at a dilution of 1:100 for 1 h at room temperature (RT). HIER made in BOND target epitope retrieval solution 2 (pH = 9). The intensity of the brown colorimetric reaction was visualized by DAB.

### 4.9. Data Analysis

Data were processed with Proteome Discoverer 2.3 (Thermo Fisher Scientific, San José, CA, USA) and searched against the Homo sapiens UniProt revised database (2018-10-01) and the B-raf V600E mutant protein sequence using the Sequest HT search engine. The search results were directly imported into Perseus software [62] to perform data normalization and filtering of missing values.

To enable comparison across the entire sample set, relative protein abundances were calculated as the ratio between the protein intensity in the sample and the intensity of the protein in the reference. The protein intensities were calculated from the TMT 11-plex reporter ions, log2–transformed and normalized by subtracting the median intensity in each sample. The relative abundances were obtained by anti-log transformation after subtracting the normalized intensity of the protein from the reference sample.

For statistical analysis, SPSS 25 (SPSS Inc, Chicago, IL, USA) was used. Two groups of tumor samples were created according to the levels of mutated B-raf (high expression *N* = 9, and low expression *N* = 7). Kaplan–Meier survival analysis with log-rank, and Breslow and Tarone Ware testing were used for univariate analysis between Groups 1 and 2. *p*-value < 0.05 was considered statistically significant and the B-raf expression values for patients reported as alive (*N* = 5) were censored.

Differentially expressed proteins between V600E_H and V600E_L were determined by Student t-test (two-tails). In this case, a *p*-value <0.01 was considered significant. PCA and heat maps were generated in R [63,64] using the packages “FactoMineR“ and “pheatmap“ respectively. For functional analysis of the differentially expressed proteins, the Ingenuity IPA Core Analysis was performed (Ingenuity, Qiagen, Germantown, MD, USA).

The availability of advanced proteomic and high-resolution nLC-MS/MS techniques has enabled the deeper mining of melanoma proteome and the determination of the relative abundance of BRAF V600E protein across several metastases. The TMT-11 plex technology used here provided multiplexing capabilities for high throughput analysis and high precision in protein quantitation. Nevertheless, as results of the huge and complex mixture of peptides infused to the mass spectrometer, a factor such as the co-isolation interference (Appendix A), i.e., multiple peptides being selected for MS/MS analysis within the same isolation window, may cause underestimation of actual protein abundances. Hence, further investigation of BRAF V600E protein in melanoma samples should include an acquisition method to improve the accuracy of protein quantitation. Future experiments should also consider the study of FFPE samples, which is the standard procedure in the clinic for melanoma tissue preservation, and the comparison of MS data with more sensitive BRAF V600E mutation detection techniques and with other protein-based assays such as immunohistochemistry.

## 5. Conclusions

The identification and relative quantitation of the B-raf V600E-mutated protein in malignant melanoma by mass spectrometry is pioneering work. To our knowledge, this is the first time that the heterogeneous translation of BRAF V600E to the level of the mutated protein in tumors from metastatic melanoma that exhibit disparate or similar expressions has been verified. This revealed a subjacent layer of heterogeneity in V600E BRAF-mutated melanomas comprised of two distinct subtypes with different molecular features and associations with different clinical outcomes, where a higher level of expression for the mutated protein was linked to a more aggressive tumor progression.

The results demonstrated the potential of proteomic techniques for molecular profiling and monitoring key mutations. Validation of our results in a larger cohort of metastatic melanoma patients will hopefully aid prognosis of the disease and improve care for melanoma patients by providing relevant information that can potentially impact treatment regimes. In conclusion, as the drugs that have been developed target the protein and not the corresponding gene, such data is mandatory clinical information.

## Figures and Tables

**Figure 1 cancers-11-01981-f001:**
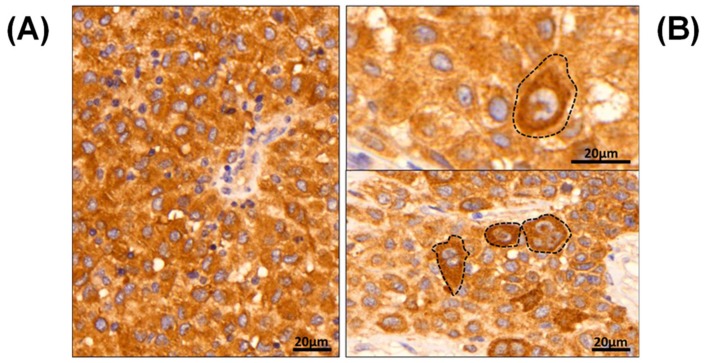
Immunohistochemical images of mutated B-raf V600E, displayed from two patients with malignant melanoma. (**A**) a patient with homogeneous B-raf expression, (**B**) two IHC images generated from two different areas of the same tumor with heterogeneous and dispersed B-raf expression, highlighted by brown colorimetric reaction. Note the grouping cells (dashed lines) with a more pronounced B-raf V600E expression pattern by brown discoloration of the HRP-DAB reaction.

**Figure 2 cancers-11-01981-f002:**
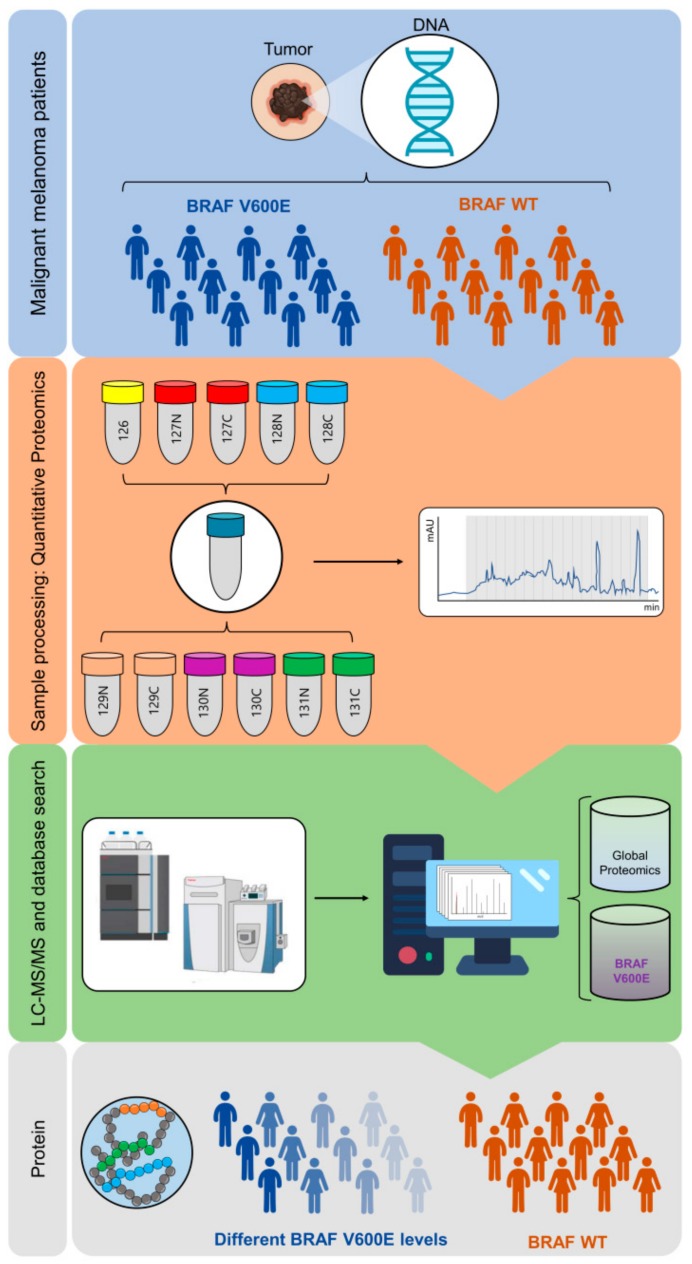
General schematic of the workflow followed for identification and quantitation of B-raf V600E from metastatic melanoma patients. Sample processing consisted of automatic generation of protein extracts from patient tumor samples, protein denaturation and tryptic digestion on a robotic micro-chromatographic platform, TMT 11-plex labelling of generated peptides, and peptide fractionation by high pH RP-HPLC. Fractions were analyzed by LC-MS/MS and the processed data searched against the human protein database (see Materials and Methods). Identification and quantitation of the B-raf V600E mutation and the proteins expressed by individual melanoma tumors (global proteomic analysis) including WT B-raf was then performed.

**Figure 3 cancers-11-01981-f003:**
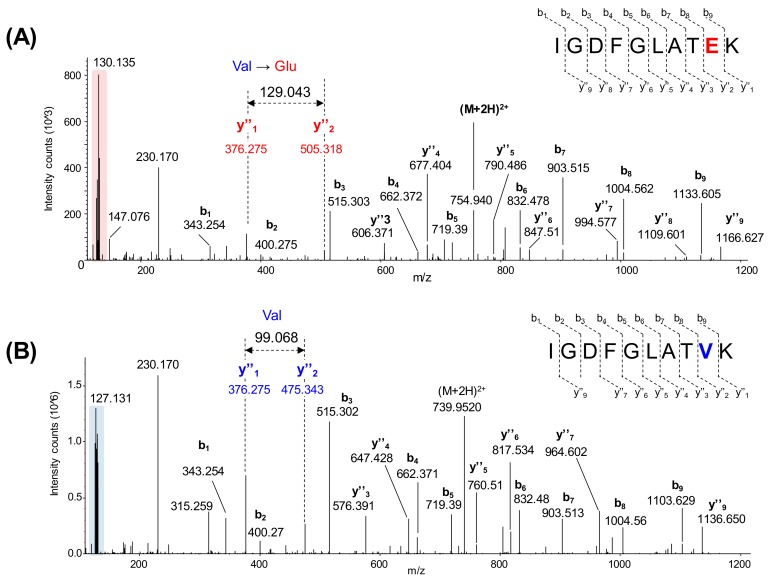
Identification of peptides from B-raf V600E and B-raf WT by mass spectrometry. (**A**) Assigned MS/MS spectrum of the TMT-labeled peptide IGDFGLATEK from B-raf V600E. Interpretation of the data showed the substitution of valine residue for glutamic acid (Glu, m/ztheo = 129.0425) that corresponds to the mutation. (**B**) Assigned MS/MS spectrum of the TMT-labeled peptide IGDFGLATVK from WT B-raf. Interpretation of the data showed the presence of the expected valine residue (Val, m/ztheo = 99.0684). The low m/z region is highlighted in both MS/MS spectra to indicate the presence of TMT 11-plex reporter ions that are used to relatively quantify the protein to provide a measure of protein expression.

**Figure 4 cancers-11-01981-f004:**
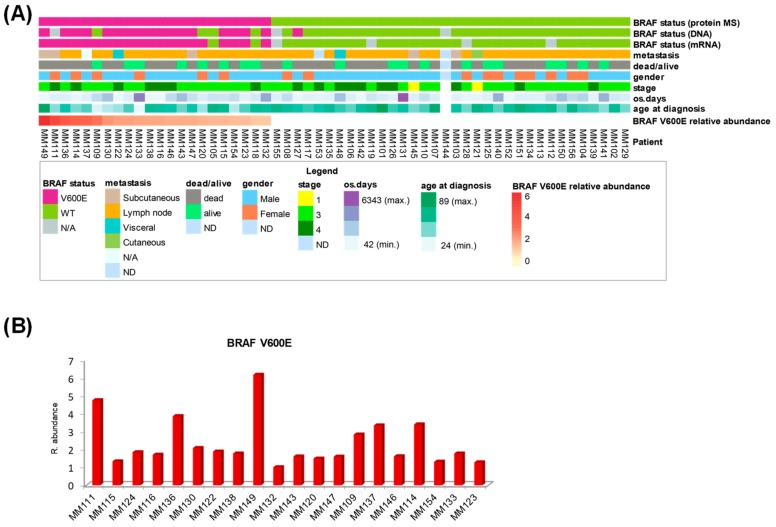
Patient clinical data and BRAF status for metastatic melanoma. (**A**) Heat map representation of patient clinical data and BRAF determination by genomic and proteomic techniques. (**B**) Plot of relative abundance of B-raf V600E protein against a reference sample.

**Figure 5 cancers-11-01981-f005:**
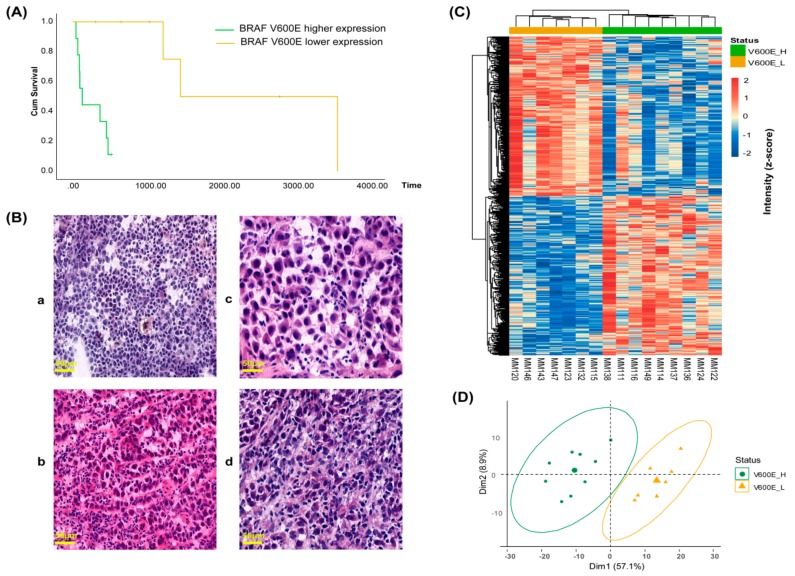
B-raf V600E expression correlated with patient survival and tumor phenotype. (**A**) Overall survival (OS) of malignant melanoma patients according to B-raf V600E mutation levels (log-rank *p* = 0.001, Breslow *p* = 0.002 and Tarone Ware *p* = 0.001). (**B**) Histological images of mutation-positive metastatic melanoma samples: (a and b) tumors MM114 and MM111 with high expression of the B-raf V600E mutated protein; and tumor (c and d) tumors MM147 and MM120 with low expression of the B-raf V600E mutated protein. For all images the magnification and scale were 10× and 50 µm, respectively. (**C**) Hierarchical clustering heat map of 697 differentially expressed proteins between the two groups of mutation-positive metastatic melanomas. (**D**) PCA of the two groups of mutation-positive metastatic melanomas based on the differentially expressed proteins. Tumor samples from each group are highlighted in common colors: high B-raf V600E expression (V600E_H, green), low B-raf V600E expression (V600E_L, yellow).

**Figure 6 cancers-11-01981-f006:**
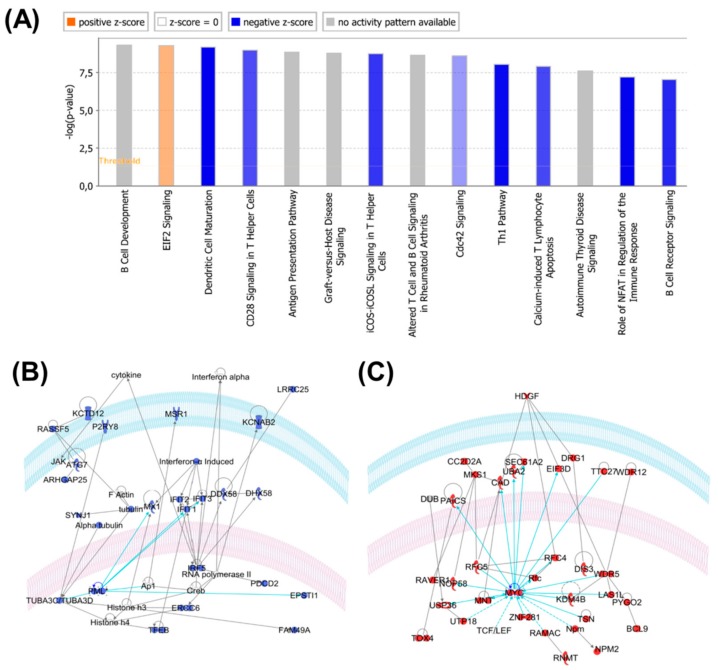
(**A**) Canonical pathways overrepresented by proteins differentially expressed in samples with low and high expressions of B-raf V600E. Blue: on average, pathway members have a lower expression of B-raf V600E_high samples. Red/orange: on average, pathway members have a higher expression of B-raf V600E_high samples. (**B**) One of the top IPA relational protein subnetworks (blue) significantly downregulated in samples with a high expression of B-raf V600E. The tumor suppressor PML is highlighted. (**C**) One of the top IPA relational protein subnetworks (red) significantly upregulated in samples with a high expression of B-raf V600E. The oncogene MYC is highlighted.

**Table 1 cancers-11-01981-t001:** Summary of the BRAF mutational status results obtained by DNA, RNA and mass spectrometry for melanoma metastases.

BRAF V600E status	Positive	Negative	Sensitivity	Specificity
DNA	18	34		
mRNA	20	32		
Protein (MS)	22	34	100%	91%

**Table 2 cancers-11-01981-t002:** Histopathological evaluation of tumors with B-raf V600E mutation.

Group	Cell Size Variation (>7 µm)	Neo-Vascularization	Discohesive Pattern	Mutli-Nucleation	Heterogeneity Score (0–4)
B-raf V600E high	5/9 (56%)	7/9 (77%)	6/9 (66%)	3/9 (33%)	3
B-raf V600E low	3/7 (43%)	1/7 (14%)	2/7 (29%)	5/7 (71%)	1

Tumors expressing high or low levels of B-raf V600E (rows) displayed heterogeneous properties with respect to tumor cohesion, vasculature, and cellular morphology (columns). The individual factors can be summed to give the heterogeneity score (final column).

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
