# Peer review of "The Hidden Story of Heterogeneous B-raf V600E Mutation Quantitative Protein Expression in Metastatic Melanoma—Association with Clinical Outcome and Tumor Phenotypes"

_cancers, 2019, doi:10.3390/cancers11121981_

Round 1

Reviewer 1 Report

Major

(1) There are only 20 positive patients in this study. A cut-off of 1.65 to distinguish BRAFV00E-high vs. BRAFV600E-low was established in this testing cohort. I would recommend testing another group of patients as a validation cohort to support their conclusion.

(2) By comparison with DNA/RNA testing, the authors established that MS can detect BRAFV600E protein. This is just a simple qualitative (i.,e., a yes or no) comparison. How did authors validate their MS assay quantitatively and determine that it can detect the quantity of BRAFV600E accurately?

(3) The heterogeneity score (0-4) in the final column of table 2 is a bit confusion. Is it a mean value of each group? Is there a standard deviation of mean? Can the author perform statistically analysis to support that BRAFV600E-high has higher heterogeneity score? The score could even carry more weight if it can be validated in a validation cohort.

Minor

(1) In line 262, it should be Figure 6 not Figure 5.

(2) Poor English from Lines 313 to 315.

Author Response

We thank the reviewer for his evaluation of the manuscript as well as for his comments and suggestions. We think the revised manuscript version has gained in presentation and clarity.

Major

(1) There are only 20 positive patients in this study. A cut-off of 1.65 to distinguish BRAFV00E-high vs. BRAFV600E-low was established in this testing cohort. I would recommend testing another group of patients as a validation cohort to support their conclusion.

Our study included altogether 56 melanoma metastases from which 20 resulted positive for BRAF V600E mutation. A cut-off of 1.65 was established from the ROC evaluation, to distinguish between low and high expression of the BRAF V600E mutation.

Further analyses linked these two groups of tumors to distinctive patient survival as well as to different tumor morphology and protein profiles.

This represents an entirely novel finding within malignant melanoma and is the major emphasis of our study outcome, i.e., to highlight that differences in mutated protein BRAF levels, will impact the patient outcome.

Overall, the study describes the group of observations we have made from the identification and quantitation by mass spectrometry of Braf V600E mutated protein in small sample set of MM tumors.

It goes beyond the objectives of this work (where the majority were lymph node metastases),  to establish the cut-off applicable to all/most melanoma metastases to distinguish low and high expression of the BRAF V600E mutation, but certainly it will be important to determine such value, as our results suggests. To achieve this goal, as the reviewer recommended, a larger cohort of metastases is needed to be analyzed.

We do not argue the comment raised by the reviewer, but it is not within the scope of this study to include a larger investigation of melanoma metastases.

Taking into account the reviewer comment and recommendation we added in the introduction on page 3, line 97 the following comments:

“...In the current work we summarize a group of observations made on small set of samples that emphasizes the importance of the analysis of B-raf V600E protein on melanoma metastases…”

Also we added in the introduction on page 3, line 112 this additional sentence:

“...Validation of our results in a larger cohort of metastatic melanoma patients will support our conclusions and could be beneficial for the treatment of melanoma patients...”

(2) By comparison with DNA/RNA testing, the authors established that MS can detect BRAFV600E protein. This is just a simple qualitative (i.,e., a yes or no) comparison.

Yes, the comparison between DNA/RNA and MS for detecting BRAFV600E was qualitative.

By MS the BRAFV600E protein was detected based on the interpretation of the MS/MS spectra of the peptide containing the mutation (figure 3A). In this way thousands of proteins are also identified in complex mixtures.

How did authors validate their MS assay quantitatively and determine that it can detect the quantity of BRAFV600E accurately?

The current TMT11 multiplex assay verifies 10 specific patient samples in relation to a control. The control sample (same in all assays), was developed to include a broad and representative melanoma metastasis disease presentation, collected from the local Southern Swedish biobank, and from n=40 patient tumors.

As result of the huge and complex mixture of peptides infused to the mass spectrometer, a factor such as the co-isolation interference i.e. multiple peptides being selected for MS/MS analysis within the same isolation window, may impact the accuracy of protein abundance determination.

We address this issue in page 12 line 391 of the current version of the manuscript

The accuracy of the MS quantitation approach (TMT 11-plex) is a challenge still, as co-isolation interference phenomenon might occur (Table 3, supplementary materials) i.e. where multiple peptides are being selected for MS/MS analysis within the same isolation window.

These effects could impact the evaluation of actual protein abundances to some extent (ref).  Hence, further investigation of BRAF V600E protein in melanoma samples should include an acquisition method that effectively eliminates the mentioned issues with TMT 11-plex quantitation.  

The extended use of the TMT technology and its ability to determine the relative abundance of proteins has been reflected in high number of publications. Below just a small selection of many examples:

A quantitative mass spectrometry-based approach to monitor the dynamics of endogenous chromatin-associated protein complexes. Nature Communications volume 9, Article number: 2311 (2018) Reproducible workflow for multiplexed deep-scale proteome and phosphoproteome analysis of tumor tissues by liquid chromatography–mass spectrometry. Nature Protocols volume 13, pages1632–1661(2018) Proteome-Wide Evaluation of Two Common Protein Quantification Methods. J Proteome Res. 2018 May 4;17(5):1934-1942. Quantitative, multiplexed workflow for deep analysis of human blood plasma and biomarker discovery by mass spectrometry. Nat Protoc. 2017 Aug;12(8):1683-1701.

In addition, previous to this study we also assessed in detail the TMT11 assay and the whole workflow using a set of 20 melanoma tumors (see poster ref) with median coefficient of variations (CV) between 4% and 16%.

(3) The heterogeneity score (0-4) in the final column of table 2 is a bit confusion. Is it a mean value of each group? Is there a standard deviation of mean? Can the author perform statistically analysis to support that BRAFV600E-high has higher heterogeneity score? The score could even carry more weight if it can be validated in a validation cohort.

Heterogeneity score in Table 2 is a score taking into account the respective histological properties. The score is a sum. To derive a yield of tissue heterogeneity, a score (0-4) was calculated and incorporated into a sum equaling these: tumor cell size variation + vascularization + discohesion + multinucleation. To accept a feature for the entire group, > 55% of the cases must display the specific property, resulting in 1. When the feature was present in less than 55%, the score was 0. Due to the low number of patients in this observational study, we did not intend to go into further analysis, which is certainly of value in a larger validation set.

Minor

(1) In line 262, it should be Figure 6 not Figure 5.

We couldn’t find this mistake, it is correct as stated.

(2) Poor English from Lines 313 to 315.

Previous version (lines 313-315 in bold)

These proteins share more than 40% sequence homology. The V600E mutation, however, is located in a region that generates a tryptic peptide that is identical for all three proteins. Therefore, quantitation of the WT B-raf with our current data and experimental design was not attempted.

Current version

These proteins share more than 40% by sequence homology. The V600E mutation, however, is located in a region that generates a tryptic peptide that is identical to all three proteins. Consequently, quantitation of the WT B-raf was not applied.

Reviewer 2 Report

In the original manuscript entitled “The Hidden Story on Heterogeneous B-raf V600E Mutation Quantitative Protein Expression in Metastatic Melanoma—Association with Clinical Outcome and Tumor Phenotypes”, Betancourt and colleagues proved that a higher level of expression for the mutated protein is associated with a more aggressive tumor progression. Overall their findings are novel, methodologically sound and clearly written. Minor points: The authors used a specific antibody targeting BRAFV600E, I suggest the authors pointed it out in the main text so that the readers would not get confused. Can’t see scale bars in Figure 5B. What is the white box in the upper right of Figure 5B a b?

Author Response

We thank the reviewer for his very positive evaluation of our work.

In the original manuscript entitled “The Hidden Story on Heterogeneous B-raf V600E Mutation Quantitative Protein Expression in Metastatic Melanoma—Association with Clinical Outcome and Tumor Phenotypes”, Betancourt and colleagues proved that a higher level of expression for the mutated protein is associated with a more aggressive tumor progression. Overall their findings are novel, methodologically sound and clearly written.

Minor points: The authors used a specific antibody targeting BRAFV600E, I suggest the authors pointed it out in the main text so that the readers would not get confused.

On page 3, line 125 the antibody targeting BRAFV600E is specified.

Can’t see scale bars in Figure 5B. What is the white box in the upper right of Figure 5B a b?

The scale bars are now visible. The white boxes were the slides themselves at loupe magnification. In the current version they were removed.

Reviewer 3 Report

Major issues:

Aim of the study is not relevant. There is huge of methods actually that quantitate BRAF mutation accurately in clinical practice and it has already been shown that BRAF VAF has been associated with prognosis and response in melanoma patients treated with a MAPK targeted therapy. There is not a “fundamental need” for more precise techniques. The title of the manuscript is not appropriate in this clinical context.

In order to validate their method, the authors compared data from sensitive mass spectrometry method to Sanger sequencing method. This latter is not the most sensitive method for genotyping, since it has a poor and weak sensitivity around 10 to 15%. Molecular sequencing methods currently widely used in clinical management and routine practice have a mutation detection cut-off around 5%, and Sanger sequencing is most of the time used in second line behind a more sensitive method. Considering this bias, the authors should bring strong validation data using a sensitive genotyping method equivalent to that used in routine.

The authors claim that the results of their study will impact the management of advanced-stage melanoma. They used fresh-frozen human metastatic tissue for validation of their method which is not the standard of care in melanoma management. Data from FFPE melanoma tissues standardly used in routine should be shown.

There is no information about the samples contain of tumor cells. This point is highly important since molecular methods have been validated when genotyping samples with at least 50% of tumor cells and below this percentage a microdissection should be performed for tumor cell enrichment.

Discordant data should be confirmed with an independent method.

Spectrometry data should be compared to IHC analysis since both methods are protein specific analysis. Indeed, previous studies compared IHC to molecular genotyping and showed strong positive and negative predictive values (100 to 97%) and (100 to 94%) respectively.

For clinical data analysis the authors “decided” to exclude 4 patients “four patients <40 years of age at diagnosis were excluded from the analysis. They argue their decision by the fact that several studies have shown higher survival below this age”. This is a strange method for performing statistical analysis. Analysis of the entire studied population should be performed.

There are some critical technical issues:

- the authors did not specify if analyzes were conducted by multiplet (minimum 3 recommended);

- "missing values" is a critical point. The authors did not specify in how many batchs they had worked: the more the number of batches is important and the more the percentage of missing values increases.

- they did not specify if they had inserted an internal standard to standardize the results of all the batches, nor a CQ common to all the batches

- information concerning "reporter ion interference" and "coelution interference" must be provided.

Author Response

We appreciate the reviewer’s critical and insightful evaluation of our work. We have tried to answered all his comments and questions to the best of our ability. 

Comments and Suggestions for Authors

Major issues:

Aim of the study is not relevant. There is huge of methods actually that quantitate BRAF mutation accurately in clinical practice

With all due respect to the reviewer, we felt that this comment is not correct.

To the best of our knowledge there are no methods in clinical practice that quantitate BRAF V600E variant at the protein level.

Most of developed methods deal with BRAF V600E mutation at DNA level.

The presence of a gene variant t DNA or transcript levels (mutated or not) does not guarantee the expression of the protein it encodes for. Even mRNA levels have low to modest correlation with protein expression due to additional post-transcription and post-translation regulations.

Insights into the regulation of protein abundance from proteomic and transcriptomic analyses. Nat. Rev. Genet.  2012; 13:227–232. Genespecific correlation of RNA and protein levels in human cells and tissues. Mol. Syst. Biol.  2016; 12:883. Can we predict protein from mRNA levels? Nature. 2017; 547:E19. Global variability analysis of mRNA and protein concentrations across and within human tissues. NAR Genomics and Bioinformatics, 2020, Vol. 2, No. 1

These are the main reasons among others which contribute to the development proteomics field and its increasing use in clinic related research.

and it has already been shown that BRAF VAF has been associated with prognosis and response in melanoma patients treated with a MAPK targeted therapy.

In the present study we look at the overall patient survival despite treatment.

We do not argue the conclusions of the mentioned study regarding BRAF V600E association with prognosis and response in melanoma patients treated with a MAPK targeted therapy.

However, as we stated in the discussion section, there has been diverse and contradictory information surrounding patient survival when analyzing both primary and metastatic melanomas, which indicated that the prognostic value of BRAF mutations at DNA level is still matter of discussion. We listed below a selection of studies that cover the spectrum of this debate.

BRAF mutation status in primary, recurrent, and metastatic malignant melanoma and its relation to histopathological parameters. Dermatol Pract Concept. 2019 Jan 31;9(1):54-62. Association between NRAS and BRAF mutational status and melanoma-specific survival among patients with higher-risk primary melanoma. JAMA Oncol. 2015, 1, 359–368. Survival according to BRAF-V600 tumor mutations - An analysis of 437 patients with primary melanoma. PLoS One 2014, 9. BRAF mutation status is an independent prognostic factor for resected stage IIIB and IIIC melanoma: implications for melanoma staging and adjuvant therapy. Eur. J. Cancer 2014, 50, 2668–76. Correlation of BRAF and NRAS mutation status with outcome, site of distant metastasis and response to chemotherapy in metastatic melanoma. Br. J. Cancer 2014, 111, 292–299. The clinical significance of BRAF and NRAS mutations in a clinic-based metastatic melanoma cohort. Br J Dermatol. 2013 Nov; 169(5):1049-55.

There is not a “fundamental need” for more precise techniques. The title of the manuscript is not appropriate in this clinical context.

This is a statement that during our decades of Melanoma Cancer treatment in hospitals covering several European countries, we have never heard from patients. On the contrary, we are very much in need of exactly novel and precise techniques, since to many patients we treat with targeted treatments are non-responders! And the heterogeneous tumor disease presentation is highly complex.

This is the first mass spectrometry study to present BRAF V600E mutation data at protein level on metastasis progression. Our study uncovers a heterogeneous expression of BRAF V600E protein in melanoma as well as differences in protein profiles of high and low BRAF V600E expressing metastases, which associate to patient survival for the two groups of metastases. None of these findings could have been achieved without looking at the BRAF V600E protein and the overall protein expression of melanoma tumors using MS and proteomic techniques. The inhibitor therapy targeting BRAF V600E mutation is developed against the protein and not against the corresponding gene. Altogether these aspects provide sufficient grounds for the need of studies like ours and we do consider the title appropriate of our manuscript.

In order to validate their method, the authors compared data from sensitive mass spectrometry method to Sanger sequencing method. This latter is not the most sensitive method for genotyping, since it has a poor and weak sensitivity around 10 to 15%. Molecular sequencing methods currently widely used in clinical management and routine practice have a mutation detection cut-off around 5%, and Sanger sequencing is most of the time used in second line behind a more sensitive method. Considering this bias, the authors should bring strong validation data using a sensitive genotyping method equivalent to that used in routine.

We agreed with the reviewer that Sanger sequencing method is not the most sensitive technique. As controls of our assay we mixed SK-MEL2 (WT BRAF cell line) with SK-MEL28 (BRAF V600E cell line) at different ratios and achieved a sensitivity of 10%.

Sanger sequencing is still chosen as a reference in several European pathology labs to detect BRAF V600E mutation. Several studies have found agreements >95% between Sanger sequencing and routine diagnostic tests such as  Cobas 4800 BRAF V600 Mutation Test and THxID™-BRAF test:

Molecular testing for BRAF mutations to inform melanoma treatment decisions: a move toward precision medicine. Modern Pathology (2018) 31, 24–38 Comparison between two widely used laboratory methods in BRAF V600 mutation detection in a large cohort of clinical samples of cutaneous melanoma metastases to the lymph nodes. Int J Clin Exp Pathol 2015;8(7):8487-8493 Comparative evaluation of the new FDA approved THxID™-BRAF test with high resolution melting and sanger sequencing. BMC Cancer 2014, 14:519

Consequently, the limitation in sensitivity of Sanger technique does not neglect the novel findings of our study:

BRAF V600E protein could be identified by mass spectrometry in melanoma metastases The protein has a heterogeneous expression that correlates with patient survival and also defined two groups of metastases. These groups of metastases are characterized by different clinical outcome, different tumor phenotypes and different protein profiles that support the differences in survival for the two groups of metastases.

As we stated in the current version of the manuscript page 3, line 97

“...In the current work we summarize a group of observations made on small set of samples that emphasizes the importance of the analysis of B-raf V600E protein on melanoma metastases…

Considering the reviewer’s comment we added in page 12, line 400 this additional sentence:

“Future experiments should also consider the study of FFPE samples, which is the standard procedure in the clinic for melanoma tissue diagnosis, the comparison of MS data with a more sensitive technique for BRAF V600E mutation detection, and also with other protein-based assays such as immunohistochemistry.”

The authors claim that the results of their study will impact the management of advanced-stage melanoma. They used fresh-frozen human metastatic tissue for validation of their method which is not the standard of care in melanoma management. Data from FFPE melanoma tissues standardly used in routine should be shown.

Protein expression analysis and quantitation is best performed on fresh tumor tissue, that is processed following our protocol, where we have research nurses within the surgery room, ensuring ultra-low temperature Biobank storage within 30 minutes after surgical isolation.

Our Biobank processing work flow ensures sample integrity and high quality, and in agreement with the Swedish Biobank law.

The reviewer is right. FFPE procedure is indeed the most widely used approach for routine tissue preservation. Nevertheless, a disadvantage of formalin fixation is generation of extensive chemical modification on biomolecules, including cross-linking between proteins and nucleic acids, hampering molecular investigation on modified species.

Utility of formaldehyde cross-linking and mass spectrometry in the study of protein–protein Interactions, J. Mass Spectrom. 2008; 43: 699–715 Characterization of histone-related chemical modifications in formalin-fixed paraffin-embedded and fresh-frozen human pancreatic cancer xenografts using LC-MS/MS. Lab Invest. 2017 Mar;97(3):279-288. doi: 10.1038/labinvest.2016.134. Epub 2016 Dec 12.

This indicates the poorer quality of FFPE tissues compared to fresh-frozen tissues, and that the later approach preserves the cells and subcellular components including nucleic acids and proteins optimally for research works.

In fact, some studies have raised concerns regarding whether extracted proteins from FFPE tissues reliably reflect their actual abundance pattern in the fresh‐frozen counterpart. (Factors that drive the increasing use of FFPE tissue in basic and translational cancer research, Biotech Histochem. 2018;93(5):373-386)

We consider the use of FFPE melanoma tissues as valuable but subsequent step in our investigation.

Considering the reviewer’s opinion we added in page 12, line 400 this sentence:

“..“Future experiments should also consider the study of FFPE samples, which is the standard procedure in the clinic for melanoma tissue diagnosis””

There is no information about the samples contain of tumor cells. This point is highly important since molecular methods have been validated when genotyping samples with at least 50% of tumor cells and below this percentage a microdissection should be performed for tumor cell enrichment.

We thank the reviewer for pointing this important aspect. We agree that content of tumor cells was missing from the manuscript. This has been added now.

Our histopathology evaluation indeed included the determination of tumor cell content. The table below shows that most BRAF V600E positive metastases had high tumor cell composition (>75%). The only exception was sample MM116 with 15% of tumor cells. This was indicative of high concentration of BRAF V600E in the tumor cells of MM116. In fact, this sample belongs to the group of high expression BRAF V600E tumors, matching the distinctive characteristics of this group such as poorer survival, tumor morphology and protein profiles.

We included in material and methods on page 13 line 420 the following:

Table 1 of supplementary  materials details patient clinical data as well as the tumor content of analyzed samples

We also added on page 12 line 374 the following comment:

Most BRAF V600E positive metastases had tumor content >75% (table 1 supplementary information). The only exception was sample MM116 with 15% of tumor cells. This was indicative of high concentration of BRAF V600E in the sample. In fact, MM116 belongs to the group of tumors with high expression of BRAF V600E, matching the distinctive characteristics of this group of metastases.

Discordant data should be confirmed with an independent method.

Two discordant results were found when comparing MS and RNA-based analyses (see table below and table 1 of suppl. information for more details). The DNA based analysis (obtained from patient records) was then used to confirm the BRAF status of the samples MM105 and MM18, which ultimately decreased the specificity of the MS analysis down to 91%. These two samples were excluded from the analyses regarding BRAF V600E.

Sample

BRAF status  (DNA)

BRAF  status (mRNA)

BRAF status  (MS)

MM105

WT

WT

V600E

MM118

WT

WT

V600E

We included the following comment in page 7 line 189

“The DNA-based analysis of the discordant cases confirmed the results of the RNA based study and the samples were excluded from the group of metastases expressing the mutation.”

Spectrometry data should be compared to IHC analysis since both methods are protein specific analysis. Indeed, previous studies compared IHC to molecular genotyping and showed strong positive and negative predictive values (100 to 97%) and (100 to 94%) respectively.

We do agree that this is an area that requires further research. Consequently, we think that the comparison of MS data with IHC is beyond the scope of the present study.

Immunohistochemistry (IHC) analysis on BRAFV600E mutated protein expression are validated and currently performed on FFPE melanoma samples. Our study however, was done using fresh-frozen melanoma tissues.

To show our case, we present here a head-to-head comparison of FFPE and Frozen IHC images from another tumor sample, as an illustration, generated by the hospital pathology standard protocols, which clearly proves the incomplete resolution of frozen IHC (see images).

Figure. Examples of IHC of FFPE (a) and fresh frozen (b, c, d) tissue samples obtained from human melanoma patients. Note the intratumoral heterogeneity of the paraffin embedded tissue sample (a) where the left part (from the dashed line) of the BRAFV600E-mutated melanoma cells displayed no immunoexpression of mutated protein whereas the right part exhibited strong immunoexpression of the mutated BRAF protein. Stromal inflammatory cells (STR) served as negative inner controls for the staining. In cryosections of BRAFV600E-mutated melanoma samples, both homogenous (b) and heterogenous (c) patterns of expression was shown (arrows indicate negative melanoma cells for the mutated protein). A fresh frozen wild type melanoma sample (d) served as a negative control for the IHC method.

Although paraffin based IHC has good predictive values in the literature, it has the following shortcomings:

the IHC resulted only yes (positive) or no (-negative) answers, it does not focus on tumor heterogeneity, therefore there are not established cut-off values the depth on protein profile information of melanoma metastases provided by the MS analysis is out of the reach of IHC

Finally, though IHC can achieve the above mentioned specificity and sensitivity values, this has not always been the case as has been shown in recent studies.

Study

Sensitivity

Specificity

Br J Biomed Sci. 2019 Apr;76(2):77-82

80.8%

100%

Melanoma Res. 2018 Apr; 28(2): 96–104

94%

95%

PLoS One. 2019 Aug 15;14(8)

76%

100%

BMC Cancer. 2016; 16: 905

89.2

96.2

Our study

100%

91%

The table suggests that this very first MS study on BRAF V600E protein expression on malignant melanoma could have comparable results with the IHC, a technique that has been around for a while. It certainly encourages developing further the MS method and performing more thorough comparison in future studies.

We included in page 12 line 400 the comment:

Future experiments should also consider the study of FFPE samples, which is the standard procedure in the clinic for melanoma tissue diagnosis, the comparison of MS data with a more sensitive technique for BRAF V600E mutation detection, and also with other protein-based assays such as immunohistochemistry..”

  7-For clinical data analysis the authors “decided” to exclude 4 patients “four patients <40 years of age at diagnosis were excluded from the analysis. They argue their decision by the fact that several studies have shown higher survival below this age”. This is a strange method for performing statistical analysis. Analysis of the entire studied population should be performed.

We acknowledge that we did not state in the manuscript all the facts driving our analysis and decision.

Our study was focused on a cohort of (20) of BRAF V600E positive tumors out of 56 samples studied. The graph below shows a markedly imbalance towards a higher overall survival of patients younger than 40 years for both, BRAF V600E positive tumor and the whole cohort, suggesting a different progression of melanoma disease for this group of patients.

Our observation was supported by similar findings in the literature. These facts led us to exclude the 4 patients < 40 years of age at diagnosis from the analysis.

In the current version of the manuscript we have adjusted our conclusions regarding the overall survival of BRAF V600E positive melanomas to patients older than 40 years old.

Also we added in page 7 and line 208 the comment.

We investigated the correlation between the relative abundance of the B-raf V600E protein and survival. We noticed a marked imbalance towards a higher overall survival of patients younger than 40 years for both, BRAF V600E positive tumor and the whole sample set (figure 2 supplementary materials), suggesting a different progression of melanoma disease for this group of patients. Several studies have also shown that survival is higher below this age [18–20]. Consequently, four patients < 40 years of age at diagnosis were excluded from the analysis.

There are some critical technical issues:

8- the authors did not specify if analyzes were conducted by multiplet (minimum 3 recommended);

Multiple or replicated analyses are recommended to reduce missing values and achieve maximal coverage in protein identification. (Reproducible quantitative proteotype data matrices for systems biology. Mol Biol Cell. 2015 Nov 5;26(22):3926-31).

Dynamic measured concentration range was improved in our study by using high pH RP-HPLC fractionation of peptides. In total 24-25 fractions were collected for each batch of TMT and analyzed by LC-MS/MS.

In addition exploratory experiments in our lab showed that duplicated and triplicated analyses of the peptide fractions had moderate impact on overall performance. Consequently, fractions were measured only once by LC-MS/MS.

The table below shows the number of proteins identified in each TMT batch, which indicate that we achieved a high coverage of the proteome of melanoma metastases. To the best of our knowledge this represents the largest number of proteins identified in melanoma and therefore the deepest proteome coverage.

TMT Batch#

1

2

3

4

5

6

Proteins identified

11329

11095

11023

11306

11204

10292

All these analyses were not included in the manuscript because it would deviate from the main focus of our manuscript, which is highlight the importance of  BRAF V600E protein expression associated with the clinical outcome, tumor morphology and protein profiles.

Following the reviewer’s comment we added in the materials and method section on page 14 line 448 the words in parenthesis

nLC-MS/MS analysis of peptide fractions (one-time per fraction) were performed on an Ultimate 3000 HPLC coupled to a Q Exactive HF-X mass spectrometer (Thermo Scientific, San Jose, CA).

9- "missing values" is a critical point. The authors did not specify in how many batches they had worked: the more the number of batches is important and the more the percentage of missing values increases.

We worked with 6 batches of TMT and identified 12210 proteins in total. This represents a high protein coverage of the melanoma proteome and also it represent what is possible to achieve with currently best performing proteomic technology.

For the 6 batches of TMT-11, 8928 proteins were commonly identified.  We did notice a decrease in the number of commonly identified proteins as the number of batches increased, which is related to the missing values.

However, it is important to notice that our work focuses on a single protein: BRAF V600E, which was identified in all TMT 11 batches. Missing values might have influenced the number of proteins that differentiates the tumor groups of low and high expression of BRAF V600E protein (697 proteins). This assumption however does not affect our conclusion regarding the differential protein expression for these two groups of metastases.

Taking into account the reviewers comments we made to the manuscript the following additions:

 in page 13 line 446:

This procedure was repeated for the six batches of TMT 11-plex analyzed.

In page 5 line 160

“More than 12000 proteins were identified and quantified with this strategy, and nearly 9000 were common to the six batches of TMT 11-plex experiments. This included the mutated B-raf V600E…”

10- they did not specify if they had inserted an internal standard to standardize the results of all the batches, nor a CQ common to all the batches

A reference sample as stated in materials and methods (page 14 page 474) was used to standardize the results between TMT-11 batches allowing to correct batch effect and  compared protein quantitation across the whole sample set of metastases:

“To enable comparison across the entire sample set, relative protein abundances were calculated as the ratio between the protein intensity in the sample and the intensity of the protein in the reference.”

The reference sample (same in all batches), was developed to include a broad and representative melanoma metastasis disease presentation, collected from the local Southern Swedish biobank, and from n=40 patient tumors.

Quality controls were also introduced to assess the performance of LC-MS/MS systems. A protein digest from Hela cells (Pierce HeLa Protein Digest Standard, Thermo Fisher Scientific) mixed with a standard peptide mixture (Pierce Peptide Retention Time Calibration Mixture) was used as QC sample and measured every 10 LC-MS/MS analysis.

Following the reviewers comments we added to the material and methods section the following text:

- on page 13 line 412:

A reference sample was developed to include a broad and representative melanoma metastasis disease presentation from n=40 patient tumors.

-Also we added in the Materials and Methods (nLC-MS/MS analysis section) of supplemental materials the following:

Quality controls were introduced to assess the performance of LC-MS/MS systems. A protein digest from Hela cells (Pierce HeLa Protein Digest Standard, Thermo Fisher Scientific) mixed with a standard peptide mixture (Pierce Peptide Retention Time Calibration Mixture) was used as a QC sample and measured every 10 LC-MS/MS analysis. This allowed monitoring peak width, retention time, base peak intensity, number of MS/MS, PSMs, number and of peptides and proteins identified, among others QC metrics.

11- information concerning "reporter ion interference" and "coelution interference" must be provided.

We attached to the supplementary materials the reporter ion interference, as is given by the manufacturer for each batch of TMT 11-plex reagents.

We also added Materials and Methods (Data analysis section) of supplementary materials the following:

The PD software allowed the introduction of reporter ion interferences for each batch of TMT 11-plex reagents as isotope correction factors in the quantification method.

And on page 13 page 439 we added

The reference sample was placed in channel 126 of the TMT 11-plex experiment in order  to reduce the effect of isotopic cross-contamination as a recent study has recommended. (Multibatch TMT Reveals False Positives, Batch Effects, and Missing Values. Mol Cell Proteomics. 2019 Oct;18(10):1967-1980).

In the current version of the manuscript the information concerning co-elution interference is provided in table 3 of supplementary materials.

Table 3. Co-isolation interference for the mutated peptide of the BRAF V600E protein in the TMT 11-plex batches

TMT Batch#

1

2

3

4

5

6

% Isolation interference

27

41

0

20

0

0

 For TMT-based quantitation, the co-elution interference has been found as the main cause for an underestimation of actual protein and peptide abundance differences, a phenomenon called ratio compression.

We applied several solutions suggested for this problem with the exception of MS3 acquisition which is not supported by the mass spectrometer used (Q Exactive HF-X):

Reduction of sample complexity entering to the MS by using high pH RP-HPLC fractionation of the peptides and 2 hour of LC-MS/MS gradient. Narrowing the isolation windows of precursor ion isolation to 0.7 Da in the setting of the MS instrument software.

Minimizing the effects of coelution interference still represents a challenge. Hence, further investigation of BRAF V600E protein in melanoma samples by MS should include an acquisition/quantitation method that effectively reduces/eliminates the mentioned issues with TMT 11-plex quantitation.

We included in page 14 line 492 the comment:

The availability of advanced proteomic and high-resolution nano-LC-MS/MS techniques has enabled the deeper mining of melanoma proteome analysis and the determination of the relative abundance of BRAF V600E protein expression across patient tumor metastases. The accuracy of the MS quantitation approach (TMT 11-plex) is a challenge still, as co-isolation interference phenomenon might occur (Table 3, supplementary materials) i.e. where multiple peptides are being selected for MS/MS analysis within the same isolation window.

These effects could impact the evaluation of actual protein abundances to some extent (ref).  Hence, further investigation of BRAF V600E protein in melanoma samples should include an acquisition method that effectively eliminates the mentioned issues with TMT 11-plex quantitation.

Reviewer 4 Report

In this manuscript the authors identified and quantified the expression of WT and mutant BRAF protein in 54 tumor tissues at the DNA, RNA and protein levels. The cohort included 19 BRAFV600 mutant, with N=9 high expression, and N=7 low expression. The statistical analysis showed that the protein levels of BRAF V600E rather than mutational status, is a bad prognostic factor for metastatic melanoma.

The data are interesting because they introduce an additional factor to evaluate melanomas.

The manuscript is repetitious and needs to be shorten

Author Response

We thank the reviewer for the encouraging support of our study.

In this manuscript the authors identified and quantified the expression of WT and mutant BRAF protein in 54 tumor tissues at the DNA, RNA and protein levels. The cohort included 19 BRAFV600 mutant, with N=9 high expression, and N=7 low expression. The statistical analysis showed that the protein levels of BRAF V600E rather than mutational status, is a bad prognostic factor for metastatic melanoma.

The data are interesting because they introduce an additional factor to evaluate melanomas.

The manuscript is repetitious and needs to be shorten

We thank the reviewer for his very positive evaluation of our work.

We do agree, and we have shortened the manuscript by removing large sections and condensing the parts that were repetitious. Most of these changes were within the discussion and conclusion parts.

Round 2

Reviewer 2 Report

The authors have addressed my concerns.

Reviewer 3 Report

The authors answered in detail to the reviewer comments.